# Diagnostic Challenges in Extrapulmonary Tuberculosis: A Single-Center Experience in a High-Resource Setting at a German Tertiary Care Center

**DOI:** 10.3390/idr17030039

**Published:** 2025-04-23

**Authors:** Jonas Wilmink, Richard Vollenberg, Ioana D. Olaru, Julia Fischer, Jonel Trebicka, Phil-Robin Tepasse

**Affiliations:** 1Department of Medicine B for Gastroenterology, Hepatology, Endocrinology and Clinical Infectiology, University Hospital Münster, 48149 Münster, Germany; jonas.wilmink@ukmuenster.de (J.W.); richard.vollenberg@ukmuenster.de (R.V.); julia.fischer2@ukmuenster.de (J.F.); jonel.trebicka@ukmuenster.de (J.T.); 2Institute of Medical Microbiology, University Hospital Münster, 48149 Münster, Germany; 3Clinical Research Department, London School of Hygiene & Tropical Medicine, London WC1E 7HT, UK

**Keywords:** extrapulmonary tuberculosis, acid-fast bacilli, diagnostic challenges, disseminated tuberculosis, complications, *Mycobacterium tuberculosis*

## Abstract

**Background/Objectives**: Extrapulmonary tuberculosis accounts for a significant portion of tuberculosis cases, presenting unique diagnostic challenges due to its heterogeneous manifestations and paucibacillary nature. This study aims to fill this gap by evaluating the diagnostic outcomes and correlations between different specimen types and test results. **Methods**: A retrospective analysis of electronic medical records of patients diagnosed with TB between January 2013 and December 2023 was carried out. The data extracted included patient demographics, comorbidities, TB classification, specimen types, microbiological test results, and time intervals to diagnosis. Statistical analysis was applied to compare the variables between pulmonary and extrapulmonary/disseminated TB groups. **Results**: Most patients were male (62.4%) and born outside of Germany (74.2%). Comorbidities, such as diabetes, cardiac disease, immunosuppressed status, and HIV, were common. Among the 194 patients, 98 had pulmonary TB, and 96 had extrapulmonary/disseminated TB. A comparison of pulmonary vs. extrapulmonary TB showed that extrapulmonary TB patients had a longer diagnostic delay (*p* = 0.013), more symptoms (*p* = 0.001), and more complications (42.7% vs. 16.3%, *p* < 0.001). Diagnostic challenges were evident, with multiple invasive procedures required in 43.5% of the extrapulmonary TB cases. **Conclusions**: This study highlights the complex clinical presentation of tuberculosis, particularly in patients with extrapulmonary and disseminated forms, who experience delayed diagnosis and more complications. These challenges in diagnosing extrapulmonary TB emphasize the need for improved diagnostic strategies and early identification, especially in high-risk populations.

## 1. Introduction

Extrapulmonary tuberculosis (EPTB) represents a significant and increasingly prevalent subset of tuberculosis (TB) cases worldwide. The World Health Organization estimates that approximately 15% of all TB cases are EPTB, with higher proportions reported in HIV-positive individuals [1]. In Europe, around a quarter of patients with TB have extrapulmonary involvement [2], with migrants, especially those from Southeast Asia and Africa, being more likely to suffer from EPTB [3,4]. In Germany, 24% of patients have EPTB, the vast majority of whom are born outside the country [5]. Despite this growing burden, EPTB remains underdiagnosed and underreported due to several unique challenges associated with its diagnosis.

The clinical presentation of EPTB is diverse and often nonspecific, making it difficult to distinguish it from other infectious or inflammatory diseases. Furthermore, the bacteriological confirmation of *Mycobacterium tuberculosis* in extrapulmonary specimens can be challenging due to their paucibacillary nature and difficulty in obtaining suitable samples [6]. Acid-fast bacilli (AFP) smear microscopy has low sensitivity for EPTB, while for nucleic acid amplification tests, like Xpert MTB/RIF and cultures, the sensitivity is variable depending on the site affected [7,8,9,10,11].

While AFB staining offers rapid results, it has poor sensitivity, ranging between 20% and 50%, depending on the sample type [11]. Culture-based methods serve as reference standards for mycobacterial infection confirmation; however, their turnaround time ranges from two to eight weeks, resulting in protracted diagnosis periods and compromised patient outcomes. Nucleic acid amplification tests (NAATs) are increasingly recognized for their superior performance characteristics, such as faster turnaround times and enhanced sensitivity, and significant progress is being made to expand their availability, even in resource-constrained settings [11,12].

Imaging techniques, such as computed tomography and magnetic resonance imaging, play crucial roles in guiding clinicians toward appropriate diagnostic workups; however, they cannot definitively confirm EPTB [13,14,15]. Lastly, biopsy and histopathology remain essential tools for diagnosing EPTB but require invasive procedures and expertise for interpretation [16,17].

The aim of this study was to describe the characteristics of patients with EPTB presenting to a tertiary care referral hospital in Germany and to highlight the challenges of diagnosing EPTB in a high-resource setting with advanced diagnostic tools, emphasizing the need for tailored approaches to improve diagnostic accuracy and patient outcomes.

## 2. Materials and Methods

This study was conducted at the University Hospital Münster, a tertiary care center in Germany. Electronic medical records from patients admitted between 2013 and 2023 with microbiologically confirmed active TB were retrieved. Data extracted from the records included demographic information, clinical details, and laboratory results related to the diagnosis and management of TB. Molecular methods used for TB detection included Xpert MTB/RIF Ultra (Cepheid, Sunnyvale, CA, USA) for native samples and an in-house nested polymerase chain reaction for formalin-fixed samples. All procedures followed institutional ethical guidelines, ensuring patient confidentiality and compliance with data protection regulations. This study received ethical approval from the local ethics committee at Muenster University Hospital (Approval number: 2020-566-f-S).

### Statistical Analyses

For categorical variables, data were presented as counts and percentages, with comparisons made using either the chi-square test or Fisher’s exact test. Continuous variables were reported as medians with interquartile ranges (IQR) or as minimum and maximum values, with comparisons performed using either the Student’s *t*-test or the Mann–Whitney U test, depending on the data distribution. All statistical tests were two-tailed, and a *p*-value of less than 0.05 was considered statistically significant. SPSS version 26 (IBM, Chicago, IL, USA) was used for all statistical analyses.

## 3. Results

### 3.1. Patients Characteristics

A total of 194 patients were identified as diagnosed with TB and included in the study. The median age was 41 years, with a range of 16 to 89 years. Among them, 121/194 (62.4%) were males, and most had been born outside Germany (74.2%). For details of the country distribution, see Appendix A.

Regarding comorbidities, diabetes mellitus affected 9.8% of patients, cardiac disease was present in 21.6%, pulmonary disease in 14.4%, HIV infection in 9.3%, replicating hepatitis C infection in 4.1% (detection of Hep. C RNA in peripheral blood), replicating hepatitis B infection in 2.6% (detection of Hep. B DNA in peripheral blood), organ transplant recipients accounted for 3.6%, hematologic disease occurred in 3.1%, and autoimmune disease was found in 8.2%. Nearly one-fifth (18.6%) received immunosuppressive therapy.

Considering the site of TB infection, isolated pulmonary involvement was observed in 50.5%, disseminated infection with more than one infection site in 21.1%, isolated lymph node involvement in 16%, isolated bone TB in 5.2%, abdominal, urogenital, intestinal, cerebral, soft tissue, and cardiovascular TB each represented less than 5.2%.

A total of 10 patients (5.2%) suffered from multi-drug-resistant tuberculosis (MDR-TB), three (1.5%) presented with pre-XDR tuberculosis, and three (1.5%) with extensively drug-resistant tuberculosis (XDR-TB). Therapy outcomes were not evaluated for 119 (61.3%) of the patients due to loss of follow-up, whereas sustained treatment success was achieved in 55 cases (28.4%) (as defined before [18]). Unfortunately, three patients (1.5%) died during therapy, and 17 (8.8%) continued receiving treatment at the study cutoff (see Table 1).

### 3.2. Clinical and Diagnostic Differences Between Isolated Pulmonary and Extrapulmonary/Disseminated Tuberculosis: A Comparative Analysis

Next, a comparative analysis was conducted between patients with isolated pulmonary tuberculosis (*n* = 98) and those with extrapulmonary/disseminated tuberculosis (*n* = 96). The median age of patients in the pulmonary group was 44.5 years, whereas it was 35 years in the extrapulmonary/disseminated group, though this difference was not statistically significant (*p* = 0.24). Fewer patients with extrapulmonary/disseminated TB were of German origin (19.7% vs. 31.6%, *p* = 0.049).

Symptomatic disease was more prevalent in the extrapulmonary/disseminated tuberculosis group (85.4%, *n* = 82) compared to the pulmonary group (65.3%, *n* = 64), with a highly significant difference (*p* = 0.001). In patients with isolated lung disease, the diagnosis was made following contact tracing in 34 cases versus 14 cases in the other group (*p* = 0.001).

General symptoms such as fever, night sweats, and weight loss were observed with similar frequencies in both groups (*p* > 0.05). However, specific symptoms varied significantly depending on the form of tuberculosis. Pulmonary tuberculosis was more frequently associated with cough (82.8% vs. 11%, *p* < 0.001), hemoptysis (10.9% vs. 0%, *p* = 0.003), and thoracic pain (15.6% vs. 2.4%, *p* = 0.005). In contrast, patients with extrapulmonary/disseminated tuberculosis more commonly presented with lymph node swelling (43.9% vs. 0%, *p* < 0.001), abdominal pain (23.2% vs. 1.6%, *p* < 0.001), bone pain (22% vs. 1.6%, *p* < 0.001), and headache (8.5% vs. 0%, *p* = 0.018) (see Appendix A).

Additionally, the median time interval from the first physician contact to tuberculosis diagnosis was longer in the extrapulmonary/disseminated group, with a range extending up to 208 weeks, compared to just 8 weeks in the pulmonary group (*p* = 0.013).

In this study, the prevalence of HIV infection was comparable between patients with pulmonary tuberculosis (10.2%) and extrapulmonary tuberculosis (8.3%) (*p* = 0.81). Median CD4^+^ cell counts were lower in extrapulmonary tuberculosis patients (126.5 cells/µL, IQR: 44.25–225.75) compared to pulmonary tuberculosis patients (264.0 cells/µL, IQR: 84.75–442.0), though this difference was not statistically significant (*p* = 0.21). Antiretroviral therapy (ART) coverage was significantly higher in pulmonary tuberculosis patients (60%) compared to those with extrapulmonary tuberculosis (12.5%) (*p* = 0.04).

Among patients with pulmonary tuberculosis, seven (7.1%) had a hepatitis C infection, with four (57.1%) showing detectable hepatitis C RNA. In the extrapulmonary tuberculosis group, five patients (5.2%) had a hepatitis C infection, and four (80%) had detectable viral RNA. Notably, none of the patients had relevant liver dysfunction.

Patients with pulmonary TB more frequently had a positive smear microscopy for acid-fast bacilli: 40.8% vs. 13.5% among those with EPTB (*p* < 0.001), and a positive *M. tuberculosis* NAAT (74.5% vs. 29.2%, *p* < 0.001). Patients with extrapulmonary/disseminated tuberculosis experienced more frequent disease-associated complications (42.7% vs. 16.3%, *p* < 0.001), including sepsis (9.4% vs. 2%, *p* = 0.03), paradoxical immune reactions (8.3% vs. 0%, *p* = 0.003), and the need for surgical interventions (11.5% vs. 0%, *p* < 0.001). For details, see Table 2.

### 3.3. Diagnostic Yield and Procedures in Extrapulmonary/Disseminated Tuberculosis

This study involved 96 patients with extrapulmonary or disseminated TB, and tissue samples were obtained from 92 (95.8%) of the cases for diagnostic purposes. Notably, in 43.5% of the patients (*n* = 40), more than one invasive procedure was required to confirm the microbiological presence of the pathogen because the TB diagnosis was not initially suspected. The sites of specimen collection were lymph nodes (*n* = 41, 44.6%), bones (*n* = 15, 16.3%), soft tissue (*n* = 10, 10.9%), colon (*n* = 5, 5.4%), bone marrow (*n* = 4, 4.3%), brain (*n* = 4, 4.3%), urine (*n* = 4, 4.3%), stool (*n* = 2, 2.1%), liver (*n* = 2, 2.2%), blood culture (*n* = 2, 2.2%), pericardium (*n* = 2, 2.2%), and ascitic fluid (*n* = 1, 1.1%).

A histologic analysis was conducted in 68 cases (73.9%) and revealed epithelioid granulomas in 59 of these cases (86.8%). NAAT on formalin-fixed specimens was performed in 55 cases (59.8%), with 44 of these (80%) showing a positive result for *M. tuberculosis* DNA.

A microscopic analysis for acid-fast bacilli on native (non-formalin-fixed) specimens was performed in 78 cases (84.8%), and acid-fast bacilli were detected in 22 cases (28.2%). Additionally, the NAAT analysis of these 78 native specimens revealed MTB DNA positivity in 59 cases (75.6%). The TB culture was positive in 56 cases (71.8%) (for details see Table 3).

### 3.4. Diagnostic Performance of Microbiological MTB Detection Methods Across Various Specimen Types

The diagnostic performance of the MTB culture, PCR on primary specimens, and microscopy on primary specimens varied across different specimen types. In lymph node biopsies (*n* = 31), the PCR demonstrated the highest positivity rate at 87.1%, followed by the culture at 77.4%, while microscopy was less sensitive with a positivity rate of 25.8%. For bone biopsies (*n* = 15), the culture showed a high positivity rate of 86.7%, whereas the PCR was positive in 60% of cases, and microscopy was only positive in 26.7%. For details, see Table 4.

## 4. Discussion

Our results indicate that the majority of patients in our tertiary care center diagnosed with TB were male, middle-aged adults who were born outside of Germany, and comorbidities, such as diabetes, cardiac disease, pulmonary disease, and immunosuppression, were common. This finding aligns with previous research, suggesting that older age [19], male gender [20,21], underlying health conditions, such as chronic lung disease [22], immunosuppression [23,24], and birth in a low-income country, increase the risk of developing active TB [25,26]. Interestingly, our data also revealed that diabetes mellitus was a frequently found comorbidity affecting close to ten percent of the patients. Previous studies have shown that diabetes significantly increases the likelihood of developing active TB [27,28]. Patients with medication-induced immunosuppression, organ transplantation, hematologic-oncologic diseases, and HIV were frequently treated at our tertiary care center. These patients are at an increased risk for tuberculosis infection and reactivation of latent TB, as immunosuppressive therapies can trigger reactivation or disease progression, with EPTB being more common. Although latent TB infection (LTBI) screening is routine before starting immunosuppression, clinicians should remain alert for TB even after screening.

Furthermore, our analysis showed that disseminated infection and extrapulmonary involvement were common among TB patients, accounting for more than half of the cases. Lymph node involvement was the most frequently observed extrapulmonary manifestation, similar to earlier observations [29,30,31,32]. However, bone and joint TB, as well as central nervous system involvement, were infrequently observed.

Our findings comparing isolated pulmonary TB and EPTB reveal notable variations in clinical presentation, diagnostic modalities, and complications. These distinctions correspond to the trends documented in previous literature. Our study found no statistically significant difference in the median age between the pulmonary and extrapulmonary TB groups, although there was a slight trend toward younger patients in the extrapulmonary/disseminated group (35 years vs. 44.5 years). These findings are consistent with previous studies that have suggested a younger age distribution in extrapulmonary TB, although some studies report a wider age range, especially in regions with a high TB burden [1]. The lower proportion of patients of German origin among those with EPTB (19.7% vs. 31.6%) is an interesting observation, potentially reflecting the higher prevalence of extrapulmonary TB in some populations, particularly in countries with a high TB incidence [30,31].

A key finding in our study was the significantly higher prevalence of symptomatic disease in the extrapulmonary/disseminated group (85.4%) compared to the pulmonary group (65.3%). This is in line with previous research that has shown that extrapulmonary TB often presents with more varied and nonspecific symptoms, leading to a higher likelihood of symptomatic disease at presentation [33,34,35]. Extrapulmonary TB can affect multiple organ systems, making it more likely to be symptomatic and harder to diagnose early, particularly in settings where TB screening is not routine [36]. Consequently, our study revealed a significant diagnostic delay observed in the extrapulmonary TB group compared to the pulmonary group. Delayed diagnosis in EPTB is well documented, as patients may present with nonspecific symptoms, or the disease may not be suspected initially, particularly in the absence of pulmonary involvement. In contrast, pulmonary TB is often diagnosed earlier due to more characteristic symptoms, such as cough, hemoptysis, and chest pain, as well as the availability of routine screening in certain populations. Our study also found that only 28.1% of patients with EPTB had concurrent pulmonary involvement. Only 14.6% of these patients were diagnosed as a result of screening. In summary, these data suggest that the manifestations of extrapulmonary tuberculosis can be easily overlooked due to the lack of screening, especially since the focus of currently available screening programs in most countries is on detecting the contagious pulmonary form of the disease [1,34,37].

Our findings also revealed that patients with extrapulmonary/disseminated TB had a significantly higher rate of disease-associated complications (42.7% vs. 16.3%, *p* < 0.001), including sepsis, paradoxical immune reactions, and the need for operative procedures. The higher frequency of TBC bacteremia in the extrapulmonary group may reflect the systemic nature of the disease, as disseminated TB can affect multiple organ systems and predispose patients to secondary infections [34]. Paradoxical immune reactions, seen in 8.3% of patients in the extrapulmonary group, are another well-documented complication, particularly in patients with disseminated or HIV-associated TB. Such immune responses often occur after the initiation of anti-TB therapy and may complicate the clinical course [38,39,40]. The need for surgical interventions in a significant number of patients also highlights the difficulties in EPTB therapy and diagnosis. Surgical interventions, such as drainage of abscesses or biopsy for histopathological evaluation, are often required to manage complications or confirm the diagnosis in EPTB, further complicating patient care. Notably, in 43.5% of the patients, more than one procedure (including biopsies, punctures, and operative procedures) was required to confirm the microbiological presence of the pathogen. Tuberculosis was not initially suspected during the first procedure in these cases, and the samples were fixed only in formalin. As a result, a second procedure was necessary to obtain native, unfixed specimens for microbiological and, particularly, cultural pathogen confirmation.

Our study shows a significantly lower proportion of patients on combined antiretroviral therapy (cART) in the extrapulmonary tuberculosis (EPTB) group (12.5% vs. 60%). This may be linked to limited healthcare access, as more patients in this group originated from non-German countries. The lower cART coverage could explain the trend toward lower CD4^+^ counts (median 126.5 vs. 264.0 cells/µL), even though this difference was not statistically significant (*p* = 0.21). Severe immunosuppression due to delayed cART initiation may have contributed to the higher occurrence of extrapulmonary and disseminated TB. These findings highlight the need for improved HIV care access, particularly for migrants from low-resource settings.

Our study employed several diagnostic techniques, each with varying yields depending on the specimen type. Histological analysis, conducted in 73.9% of the cases, identified epithelioid granulomas in 86.8% of the samples. Although granulomas are suggestive of tuberculosis, they are not pathognomonic, as they can also be found in other conditions, such as sarcoidosis or fungal infections. Nonetheless, granulomas remain a key diagnostic clue in extrapulmonary TB, warranting further microbiological testing. NAAT analysis on formalin-fixed specimens yielded an 80% positivity for MTB DNA, confirming the utility of this method as a sensitive and rapid tool for detecting TB, even in suboptimal samples. However, the variability of NAAT across specimen types—such as formalin-fixed versus native samples—highlights its limitations in complex cases like EPTB. Microscopy was performed in 84.8% of the native specimens, with AFB detected in only 28.2%. This low yield is consistent with the generally lower bacterial load in EPTB compared to pulmonary TB, reducing the sensitivity of microscopy [41]. Unsurprisingly, microscopy had a low yield, while culture and NAAT were positive in more than 70% of the patients [41]. Finally, culture positivity was 71.8% in the native specimens, reinforcing the value of culture as the gold standard for TB diagnosis. While culture remains definitive, its time-consuming nature emphasizes the importance of using faster adjuncts like PCR and histology, especially in cases of suspected disseminated TB. Different native tissue samples exhibit varying sensitivities for tuberculosis detection via PCR. Our study reveals notable variations in TB culture positivity across specimen types: lymph node biopsies showed a 77.4% positivity rate, and bone biopsies showed 86.7%. Although other specimen types also yielded positive results, small sample sizes suggest a cautious interpretation of generalizability. Different native tissue samples exhibit varying sensitivities for tuberculosis detection via NAAT [11,42,43]. These findings underscore the potential of AFB and PCR as early indicators of active TB, especially where rapid culture results are unavailable.

One limitation of our study is that our institution serves as a major referral center for tuberculosis cases in Münster and the Münsterland region in West-Germany, receiving patients not only from local healthcare providers but also from primary reception facilities for migrants. Given its role as a specialized center, our cohort may not fully reflect the TB epidemiology of the general population but rather a subset of patients with more complex or severe presentations. We will include a description of this in the Methods section to clarify the study’s scope and applicability. A further limitation of our retrospective analysis is the potential use of different diagnostic and clinical assessment protocols for extrapulmonary TB (EPTB). Since the data collection spanned an extended period, institutional or temporal variations in diagnostic approaches and treatment strategies may have influenced the results, potentially limiting their comparability. Prospective studies with standardized assessment protocols would be necessary to provide a more precise evaluation of EPTB cases. Furthermore, due to the retrospective nature of our study, we were unable to systematically assess patient adherence to TB therapy or long-term outcomes after treatment completion. This limits our ability to evaluate treatment success and the risk of recurrence. Additionally, there is a risk of incomplete or missing data, particularly regarding follow-up documentation. A significant proportion of our cohort consisted of migrants who were initially referred to our tertiary care center from primary reception facilities. Many of these patients were later transferred to other healthcare institutions within Germany or repatriated to their home countries, making long-term follow-up challenging. Due to administrative and logistical barriers, including changes in residence status and limited access to updated contact information, continued monitoring was often not feasible. Future studies should explore strategies to improve patient tracking and inter-institutional communication to minimize losses to follow-up in similar populations.

Our results indicate that, despite a high-resource setting, optimizing local Standard Operating Procedures (SOPs) and targeted staff training could significantly improve TB care. Strengthening institutional workflows and increasing awareness among healthcare workers are key factors in order to avoid diagnostic delays, unnecessary multiple interventions for sample collection, and improving treatment outcomes. Based on our findings, we propose the following locally focused measures:
Enhanced Diagnostic Strategies:

Ensure the implementation of hospital-wide screening protocols for high-risk patients, particularly those undergoing immunosuppressive therapy.

Standardize specimen handling procedures through internal SOPs and staff workshops to ensure optimal microbiological diagnostics (e.g., avoiding formalin fixation for cultures).

Promote the routine use of NAAT and imaging (CT/MRI) for early EPTB detection, supported by radiology training sessions on TB-specific findings.
Improved Treatment Approaches:

Establish interdisciplinary case discussions (infectious diseases, pulmonology, radiology, and surgery) to optimize decision-making in TB management.

Strengthen internal hospital protocols for early cART initiation in HIV patients through close cooperation between infectious disease specialists and primary care providers.

Develop training modules for clinicians on recognizing and managing surgical indications in EPTB cases.
Strengthening Follow-up:

Improve coordination with local public health authorities and primary care physicians through structured communication pathways and regular feedback meetings.

## 5. Conclusions

Our study highlights the challenges in diagnosing and managing TB, particularly extrapulmonary and disseminated forms, even in a high-resource setting. Despite advanced diagnostic tools, delays in detection and treatment remain a concern, emphasizing the need for optimized local Standard Operating Procedures. A structured, hospital-based approach, including enhanced screening, standardized diagnostic workflows, continuous staff training, and interdisciplinary case discussions, can significantly improve TB care. Ensuring closer collaboration with public health authorities, primary care providers, and external healthcare facilities will be essential in preventing losses to follow-up and improving long-term patient management, particularly in vulnerable patient populations.

## Figures and Tables

**Table 1 idr-17-00039-t001:** Patient’s characteristics: MDR = multi-drug resistant, XDR = extended drug resistant; ** for details, see Appendix A.

Patient Characteristics (*n* = 194)		
General characteristics	Age, years, median (min, max)	41 (16–89)
	Gender, male, total (%)	121 (62.4%)
Country of origin, absolute (%)	Germany	50 (25.8)
	Other than Germany **	144 (74.2)
WHO geographic regions of origin	Europe	93 (47.9%)
	Africa	34 (17.5%)
	Eastern Mediterranean	32 (16.5%)
	Southeast Asia	18 (9.3)
	Americas	2 (1%)
	Unknown	15 (7.7%)
Pre-existing conditions, absolute (%)	Diabetes mellitus	19 (9.8%)
	Cardiac disease	42 (21.6%)
	Pulmonary disease	28 (14.4%)
	HIV infection	18 (9.3)
	Replicating hepatitis C infection	8 (4.1)
	Replicating hepatitis B infection	5 (2.6)
	Solid organ transplantation	7 (3.6)
	Hematologic disease	6 (3.1)
	Autoimmune disease	16 (8.2)
	Immunosuppressive therapy	36 (18.6)
Site of TB infection, absolute (%)	Isolated pulmonary manifestation	98 (50.5%)
	Disseminated infection (more than one site of inception)	41 (21.1%)
	Lymph node	31 (16%)
	Bone and joint	10 (5.2%)
	Abdominal	5 (2.6%)
	Urogenital	2 (1%)
	Intestinal	2 (1%)
	Cardiac	2 (1%)
	Soft tissue	2 (1%)
	Cerebral/meningitis	1 (0.5%)
Tuberculosis drug resistance, absolute (%)	MDR tuberculosis	10 (5.2)
	Pre-XDR tuberculosis	3 (1.5)
	XDR tuberculosis	3 (1.5)
Therapy outcome	Not evaluated (lost to follow-up)	119 (61.3)
	Sustained treatment success	55 (28.4%)
	Deceased	3 (1.5%)
	Therapy ongoing	17 (8.8%)

**Table 2 idr-17-00039-t002:** Comparative analysis between patients with isolated pulmonary tuberculosis and those with extrapulmonary/disseminated tuberculosis.

	Isolated Pulmonary Tuberculosis (*n* = 98)	Extrapulmonary/Disseminated Tuberculosis (*n* = 96)	*p*-Value
Age, years, median (min–max)	44.5 (16–85)	35 (18–89)	0.24
Gender, male, total (%)	66 (67.3%)	55 (57.3%)	0.15
Country of origin Germany, absolute (%)	31 (31.6%)	19 (19.7%)	0.049
Symptomatic disease	64 (65.3%)	82 (85.4%)	0.001
Diagnosed through contact tracing	34 (34.7%)	14 (14.6%)	0.001
Time interval from first physician contact until TB diagnosis (weeks), median (min–max)	1 (1–8)	1 (1–208)	0.013
Detection of drug resistance (PCR and/or culture)	16 (16.3%)	18 (18.8%)	0.66
TB diagnosis (Smear)	98 (100%)	96 (100%)	n.d.
Microscopy positive	40 (40.8%)	13 (13.5%)	<0.001
Positive MTB NAAT	73 (74.5%)	28 (29.2%)	<0.001
Culture positive	68 (69.4%)	25 (26.0%)	<0.001
Disease-associated complications	16 (16.3%)	41 (42.7%)	<0.001
Hepatotoxic side effect of tuberculostatic medication	4 (4.1%)	10 (10.4%)	0.1
Hemoptysis	2 (2%)	0 (0%)	0.5
Pancreatitis	0 0%)	2 (2.1%)	0.24
Ileus	0 (0%)	1 (1%)	0.5
TBC bacteriaemia	2 (2%)	9 (9.4%)	0.03
Paradoxical immune reaction	0 (0%)	8 (8.3%)	0.003
Required surgery	0 (0%)	11 (11.5%)	<0.001
Hemophagocytic lymphohistiocytosis	0 (0%)	2 (2.1%)	0.24
Organ transplantation	5 (5.1)	2 (2.1%)	0.45
Hematologic disease	4 (4.1%)	2 (2.1%)	0.68
Autoimmune disease	8 (8.2%)	8 (8.3%)	1
Immunosuppressive therapy	18 (18.4)	18 (18.8)	0.95
HIV infection	10 (10.2%)	8 (8.3%)	0.81
CD4^+^ cell count (cells/µL) (IQR)	264.0 (84.75–442.0)	126.5 (44.25–225.75)	0.21
Antiretroviral therapy	6 (60%)	1 (12.5%)	0.04
Hep. C infection	7 (7.1%)	5 (5.2%)	n.d.
Hep. C RNA detectable	4 (57.1%)	4 (80%)	n.d.
Hep. B infection	8 (8.2%)	12 (12.5%)	n.d.
Hep. B DNA detectable	1 (12.5%)	4 (33%)	n.d.
Drug resistance	16 (16.3%)	18 (18.8%)	0.66
TBC medical treatment	95 (96.9%)	93 (96.9)	1.0
Standard treatment (Rif, Iso, PZA, Etb)	74 (75.5%)	71 (74%)	0.8
Therapy outcome			0.24
Not evaluated (Lost to follow-up)	65 (66.3%)	53 (55.2%)	
Sustained Treatment Success	23 (23.5%)	32 (33.3%)	
Deceased	3 (3.1%)	1 (1%)	
Therapy ongoing	7 (7.1%)	10 (10.4%)	

**Table 3 idr-17-00039-t003:** Table summarizing diagnostic tests and outcomes for extrapulmonary/disseminated tuberculosis (*n* = 92), including multiple procedural requirements, biopsy distribution, histology, TBC culture, and NAAT on formalin-fixed and native specimens.

	Extrapulmonary/Disseminated Tuberculosis (*n* = 92)
More than one procedure is required for microbiological pathogen confirmation	40 (43.5%)
Histological analysis	68 (73.9%)
Detection of epithelioid granulomas	59 (86.8%)
NAAT on formalin-fixed specimens	55 (59.8%)
NAAT positive for MTB DNA	44 (80%)
TB culture done	78 (84.8%)
TBC culture positive	56 (71.8%)
Microscopic analysis on native (no formalin fixation) specimens	78 (84.8%)
Detection of acid-fast bacilli	22 (28.2%)
NAAT on native (no formalin fixation) specimens	78 (84.8%)
NAAT positive for MTB DNA	59 (75.6%)

**Table 4 idr-17-00039-t004:** Comparison of tuberculosis (TBC) culture status between various specimen types in EPTB patients.

Specimen	MTB Culture Positive	PCR Primary Specimen Positive	Microscopy Primary Specimen Positive
Lymph node (*n* = 31)	24 (77.4%)	27 (87.1%)	8 (25.8%)
Bone biopsy (*n* = 15)	13 (86.7%)	9 (60%)	4 (26.7%)
Soft tissue biopsy (*n* = 8)	4 (50%)	8 (100%)	4 (50%)
Colon biopsy (*n* = 5)	3 (60%)	4 (80%)	2 (40%)
Bone marrow biopsy (*n* = 2)	1 (50%)	0 (0%)	0 (0%)
Brain biopsy (*n* = 4)	1 (25%)	2 (50%)	0 (0%)
Urine (*n* = 4)	4 (100%)	4 (100%)	3 (75%)
Stool (*n* = 2)	0 (0%)	1 (50%)	0 (0%)
Liver biopsy (*n* = 2)	1 (50%)	0 (0%)	0 (0%)
Ascites (*n* = 1)	1 (100%)	1 (100%)	0 (0%)
Blood culture (*n* = 2)	2 (100%)	2 (100%)	0 (0%)
Pericard (*n* = 2)	2 (100%)	1 (50%)	0 (0%)

## Data Availability

The data are unavailable due to patients’ privacy.

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
