# Peer review of "Diagnostic Challenges in Extrapulmonary Tuberculosis: A Single-Center Experience in a High-Resource Setting at a German Tertiary Care Center"

_2036-7449, 2025, doi:10.3390/idr17030039_

Round 1
Reviewer 1 Report
Comments and Suggestions for Authors
Dear Editor
In this paper, the authors address the issue of diagnostic challenges in cases of extrapulmonary tuberculosis. The study is interesting and analyzes the phenomenon with clarity and accuracy.
However, there are some improvements that can be made to the paper before publication.
- Page 3, lines 91-96:HIV is a co-infection, not a condition. It is well known that patients with HIV more frequently develop extrapulmonary forms of tuberculosis. However, the clinical presentation depends on the patient's immune competence (CD4 count). Therefore, the average CD4 lymphocyte count in HIV-positive patients should be included, along with information on whether they were treatment-naïve or already receiving ART.
- RegardingHCV-positive patients, specify whether they had undergone antiviral treatment or not, and whether they had any liver function impairment.
- Line 99: bone TB
- Tabel 1: Include the WHO geographic regions of origin in Table 1 (e.g., Europe, Africa, etc.).Modify HIV as per the previous comment. Consider defining disseminated tuberculosis as the involvement of at least three organs. The treatment outcome data are extremely low—could there be an error in the calculation?
- Introduce symptoms in the table
- Table 3:The data on diagnostic procedures can be omitted and described in the text. Keep only the microbiological data in the table and specify the NAAT methods
The discussion needs improvement:
The reported case series has the following key characteristics:
- High HIV infection rateand immunouppression
- Approximately half of the patients had extrapulmonary TB
- Low cure rates and high loss-to-follow-up rates
- Discuss theincidence of extrapulmonary TB in relation to the patients' geographic origin, referencing:
Rachwal N, Idris R, Dreyer V, Richter E, Wichelhaus TA, Niemann S, Wetzstein N, Götsch U. Pathogen and host determinants of extrapulmonary tuberculosis among 1035 patients in Frankfurt am Main, Germany, 2008-2023. Clin Microbiol Infect. 2024 Nov 9:S1198-743X(24)00535-4. doi: 10.1016/j.cmi.2024.11.009. Epub ahead of print. PM
Would it be possible to introduce an analysis of diagnostic delay in the two populations?
Reviewer 2 Report
Comments and Suggestions for Authors
- Introduction can be improved with epidemilogical data of EPTB from Germany and from Europe.
- In the results, table 1 and 2 could be merged into a single table, making faster and smoother the reading.
- In table 1, specify the country (or the area) of origin for patients born outside Germany.
- In the discussione, please add an explanation of the high number of patients lost to follow-up, analyzing the reasons and discussing possible improvement strategies
- A limitation of the study is the low availability of outcome data (65% and 53% of cases of pulmonary and extrapulmonary TB, respectively) which do not allow to evaluate any differences between the two groups. Discuss this data among the limitations of the study.
Reviewer 3 Report
Comments and Suggestions for Authors
This article on extrapulmonary tuberculosis (EPTB) is of interest because it was developed in a university service in a developed low TB burden country.
Although the contribution to knowledge on the subject is limited, the comparison between pulmonary TB and EPTB shows differences that may be of local interest. For example: the majority of patients are of foreign origin (non-German); presentations with severe and complex forms (requiring surgery) and associated with comorbidities in EPTB. On the other hand, it reiterates the diagnostic difficulty that leads to delayed diagnosis in the forms of EPTB, as well as the elevated microbiological positivity in pulmonary TB.
This is a retrospective study, with data from medical records. In Methods it would be convenient to describe the representativeness of this service in the city/region of Germany where it is located. Also, which NAAT test (PCR) was used?
The Results are well presented. In the discussion, it would be important to clarify the high percentage of loss to follow-up of treatment (are patients followed up in another service?) and the limitations of a retrospective study, in which there were probably (?) different assessment protocols for the forms of EPTB.
Round 2
Reviewer 1 Report
Comments and Suggestions for Authors
Dear Editor,
I recommend the paper for publication
Reviewer 2 Report
Comments and Suggestions for Authors
No commments for authors